# Nanoscale cooperative adsorption for materials control

Rong Ye [1,4], Ming Zhao [1,4], Xianwen Mao [1], Zhaohong Wang [1], Diego A. Garzón [1,2], Heting Pu [1,3], Zhiheng Zhao[1] & Peng Chen [1✉]

Adsorption plays vital roles in many processes including catalysis, sensing, and nanomaterials design. However, quantifying molecular adsorption, especially at the nanoscale, is challenging, hindering the exploration of its utilization on nanomaterials that possess heterogeneity across different length scales. Here we map the adsorption of nonfluorescent small molecule/ion and polymer ligands on gold nanoparticles of various morphologies in situ under ambient solution conditions, in which these ligands are critical for the particles' physiochemical properties. We differentiate at nanometer resolution their adsorption affinities among different sites on the same nanoparticle and uncover positive/negative adsorption cooperativity, both essential for understanding adsorbate-surface interactions. Considering the surface density of adsorbed ligands, we further discover crossover behaviors of ligand adsorption between different particle facets, leading to a strategy and its implementation in facet-controlled synthesis of colloidal metal nanoparticles by merely tuning the concentration of a single ligand.

---

[1] Department of Chemistry and Chemical Biology, Cornell University, Ithaca, NY, USA. [2] Present address: Departamento de Química, Universidad de Los Andes, Bogotá, Colombia. [3] Present address: Department of Chemistry and Biochemistry, University of California, Los Angeles, Los Angeles, CA, USA. [4] These authors contributed equally: Rong Ye, Ming Zhao. ✉email: pc252@cornell.edu

Adsorption plays vital roles in many processes in daily life, research, and industry applications, e.g., in purification, separation, or decontamination via differentiated adsorption affinities or kinetics; in catalysis for reactant activation or catalyst poisoning; and in sensing via adsorption-induced physicochemical changes[1–6]. On nanoparticles, adsorption of molecules can stabilize their solution dispersion, control their morphology during synthesis, enhance their surface functionality, or limit their catalytic performance[3–6]. To improve these applications or explore new ones, it is essential to understand quantitatively the adsorption behaviors of molecules on surfaces. However, many aspects make it challenging, including the differentiation between adsorbed and free molecules, the often-miniature amounts of adsorbates, the interference from solvents, and the inhomogeneities of the adsorbent surfaces across different length scales. For adsorption on nanoparticles, their small sizes, multiplicity of surface facets, and intrinsic heterogeneity present further challenges, for which high-resolution, high-sensitivity, and quantitative measurements are needed[7–9], whereas traditional bulk measurements average over many particles, masking adsorption differences at single-particle or subparticle level[10]. Here, using COMPEITS (COMPetition-Enabled Imaging Technique with Super-resolution) that is capable of imaging non-fluorescent surface processes in situ and at nanometer resolution[11], we map the adsorption of small molecule/ion and polymer ligands on individual gold (Au) nanoparticles of various morphologies under ambient solution conditions (Fig. 1a). These ligands play critical roles in the shape-controlled synthesis, solution stabilization, surface functionalization, and catalytic poisoning of nanoparticles of various compositions[3–6,12]. We quantify their adsorption affinity and uncover positive/negative adsorption cooperativity, both of which can even differ among different sites on the same nanoparticle. We further discover crossover behaviors of ligand adsorption between different nanoparticle facets, leading to a strategy and its implementation in facet-tuned synthesis of colloidal metal nanoparticles.

COMPEITS[11] is based on competitive adsorption that suppresses the rate of a surface-catalyzed fluorogenic auxiliary reaction, whose products are imaged, counted, and localized with nanometer precision via single-molecule fluorescence microscopy[13–15] (Fig. 1a, Supplementary Information section 1.7). COMPEITS not only has super-optical resolution (down to ~10 nm; Supplementary Information section 3.1 and Supplementary Fig. 10b) but also probes specifically the first-layer surface adsorption of the competitor because multi-layer adsorption provides no further suppression of the auxiliary reaction rate. We used the catalytic reduction of resazurin (R) by $NH_2OH$ to fluorescent resorufin in aqueous solutions as the auxiliary reaction (Supplementary Fig. 3a), which follows the Langmuir–Hinshelwood mechanism on Au particle surfaces[14]. In the presence of excess $NH_2OH$ and a competing ligand (L), the (specific) reaction rate ($v_R$) of the auxiliary reaction follows (Eq. S11):

$$v_R = \frac{k_R K_R [R]}{1 + K_R [R] + (K_L [L])^h} \tag{1}$$

Here $k_R$ is a (specific) rate constant; $K_R$ and $K_L$ are the adsorption equilibrium constants of R and L, respectively; and $h = 1$ when L follows Langmuir adsorption. If L adsorption is cooperative, $h$ deviates from 1 and is the Hill coefficient of cooperativity: $h > 1$ for positive cooperativity; $h < 1$ for negative cooperativity[16,17].

## Results

### Cooperative ligand adsorption on single nanoparticles. We chose to study Au nanoparticles for their wide applications as catalysts or probes and for their availability in variable sizes and

shapes[18–20]. We first examined the adsorption of cetyltrimethylammonium bromide (CTAB), a ligand widely used in Au nanoparticle synthesis and surface modification[5,18], on pseudospherical 5-nm Au nanoparticles using COMPEITS. The fluorogenic reaction rate $v_R$ on a single nanoparticle decreases with increasing [CTAB] (up to 5 μM, below CTAB's critical micelle concentration of 0.96 mM at 25 °C[21]) (Fig. 1b), confirming CTAB's competition with resazurin adsorption on the nanoparticle. Control experiments show that this decrease is not due to catalyst deactivation (Supplementary Information section 3.3) nor to displacement of the excess co-reactant $NH_2OH$ (Supplementary Fig. 9m). Equation (1) satisfactorily fits the $v_R$-vs-[CTAB] titration, giving $K_{CTAB} = 0.80 \pm 0.03$ μM$^{-1}$ for this nanoparticle (Fig. 1b, red line). ($k_R$ and $K_R$ were determined in earlier titration of [R] on the same nanoparticle without CTAB; Supplementary Fig. 10k). Strikingly, the extracted Hill coefficient $h_{CTAB}$ is $2.2 \pm 0.1$ (Fig. 1e), significantly greater than 1, indicating positive cooperativity in the first-layer CTAB adsorption on the particle surface, whereas forcing $h_{CTAB} = 1$ cannot fit the data (Fig. 1b, blue line). This cooperativity is also apparent in the linear Hill plot of the data, with the slope being $h_{CTAB}$ ~2.2 (Fig. 1b, inset), reflecting that ~2 CTAB molecules adsorb concurrently. This positive adsorption cooperativity, important for high adsorption efficacy[1,6,22] and revealed here on nanoparticles, possibly stems from attractive hydrophobic interactions between the alkyl chains of CTA$^+$, as in self-assembled monolayers[23] (Br$^-$ contribution is small; see later).

To achieve statistical significance, we studied 50 nanoparticles. On average, $K_{CTAB}$ ~0.65 μM$^{-1}$; more important, they all show positive cooperativity (gray points, Fig. 1b), with the Hill coefficient $h_{CTAB}$ ~2.0 (Fig. 1e); and the particle-averaged behavior agrees with bulk measurements (Supplementary Fig. 9a). Notably, with increasing $K_{CTAB}$, $h_{CTAB}$ of individual particles approaches 1 (Fig. 1i), consistent with that stronger adsorbate–surface interactions should lead to weaker adsorbate–adsorbate interactions[1] and thus weaker cooperativity. We further studied cetyltrimethylammonium hydroxide (CTAOH) and cetyltrimethylammonium chloride (CTAC) to test the generality of positive adsorption cooperativity. Their adsorptions on 5-nm nanoparticles follow $K_{CTAB} < K_{CTAOH} < K_{CTAC}$ in affinity and they both show positive cooperativity ($h > 1$) like CTAB (Fig. 1f; Supplementary Information section 3.4).

We next studied the adsorption on 5-nm Au nanoparticles of poly-$N$-vinylpyrrolidone (PVP). Compared with CTAB/CTAOH/CTAC, PVP is a polymer with a much larger molecular weight, more flexible in molecular conformation, and also commonly used in shape-controlled nanoparticle synthesis[24]. For PVP55k (average molecular weight ~55k g/mol), its competitive adsorption is clear in single-particle reaction rate titrations (Fig. 1c). On average, $K_{PVP55k} = 2.4 \pm 0.3$ nM$^{-1}$ (Fig. 1e). More strikingly, its Hill coefficient $h_{PVP55k} = 0.75 \pm 0.09$ (Fig. 1e), clearly smaller than 1, demonstrating PVP55k's negative adsorption cooperativity, unseen previously for synthetic polymers and particularly on nanoparticles. This negative cooperativity possibly stems from direct electrostatic repulsions between positively charged PVP chains at pH ~7.4 ($N$-alkylpyrrolidone has a p$K_b$ of ~3.5)[25] or from that hydrophilic PVP chains prefer interactions with water than with themselves[24]. Among 36 nanoparticles, with increasing $K_{PVP55k}$, $h_{PVP55k}$ approaches 1 (Fig. 1j), corroborating that stronger adsorbate–surface interactions are associated with weaker cooperativity. Moreover, PVP's adsorption affinity decreases with smaller molecular weight (Fig. 1g), suggesting its multivalency-enhanced adsorption, whereas its Hill coefficient stays at ~0.7, suggesting that their inter-chain interactions are dominated by sub-chain structures, like thermal blobs, which are similar regardless of molecular weights[26].

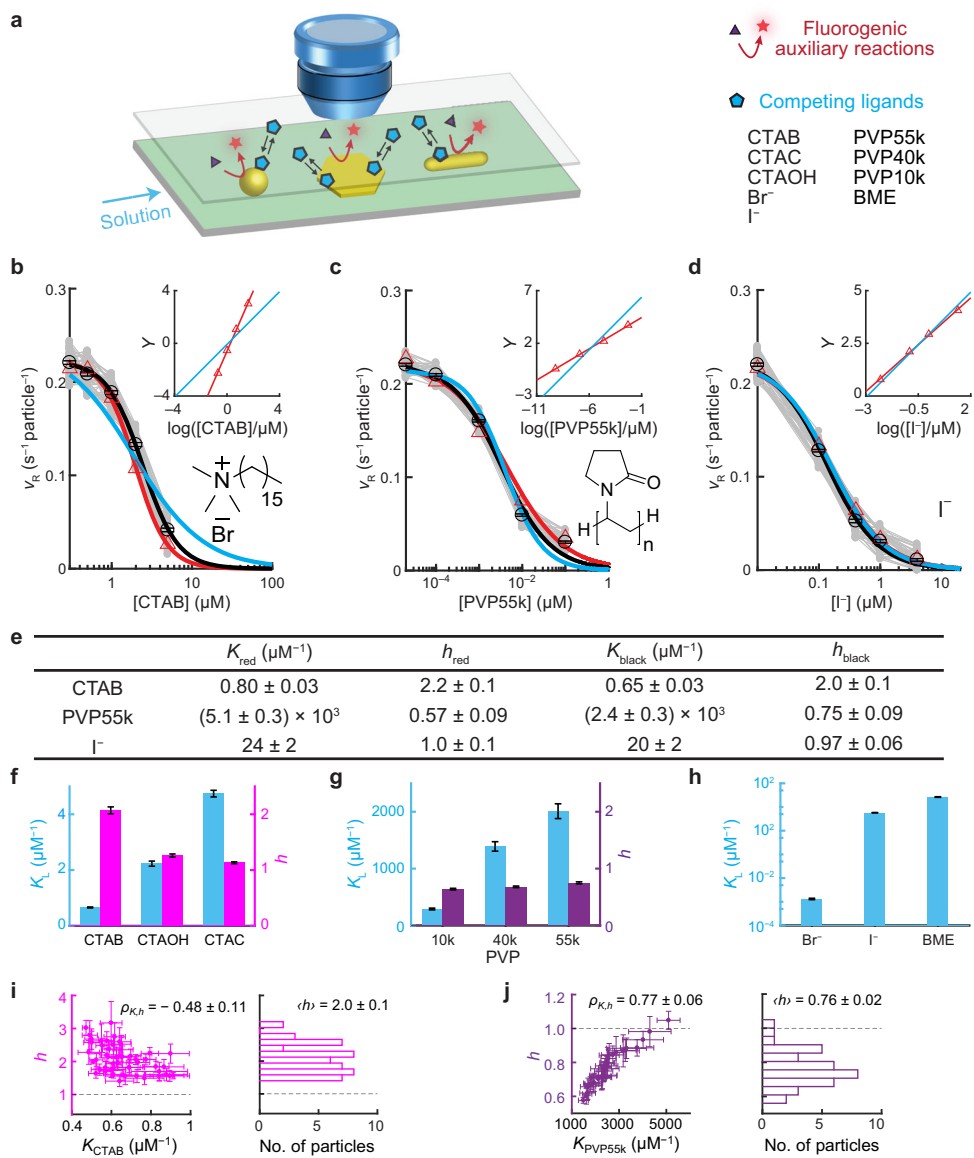

**Fig. 1 COMPEITS imaging of cooperative ligand adsorption on single 5-nm Au nanoparticles. a** Schematic of the experimental design and scopes of particles and ligands. Fluorescence is excited via total-internal-reflection geometry (Supplementary Fig. 4). **b**–**d** Single-particle titration of fluorogenic auxiliary reaction rate $v_R$ vs. [L] of 50, 36, and 44 particles for CTAB (**b**), PVP55k (**c**), and I⁻ (**d**), respectively (gray). Data points at [L] = 0 are placed on the y-axes manually. Red triangles: representative single-particle examples. Black circles: averages among particles. Red/black lines: corresponding fits with Eq. (1). Blue lines: Fits with $h$ set to 1. Insets: the corresponding Hill plots of the representative single particles (points); lines: fits with the rearranged linear Hill form of Eq. (1) (Eq. S14) with $h$ floating (red) or set to 1 (blue); the slope here is $h$. All fitting parameters summarized in **e** and Supplementary Table 2b. **e** Selected fitting parameters from **b** to **d**. **f**–**h** Particle-averaged adsorption equilibrium constants $K$ (blue) and Hill coefficients $h$ (magenta/purple) of CTA⁺ with different counter-anions (**f**), PVP with different molecular weights (**g**), and ligands showing no cooperativity (**h**). Distributions among individual particles are in Supplementary Fig. 12j–n. **i**, **j** Left: $h$ vs. $K$ for CTAB (**i**) and PVP55k (**j**); each point is from one nanoparticle; $\rho_{K,h}$: Pearson's cross-correlation coefficient (see the definition in Supplementary Information section 4.3). Right: histograms of $h$. Error bars are s.e.m. in **b**–**h** for comparing the mean values and s.d. in **i**, **j** to show the uncertainty of the fitted parameters.

Considering CTA⁺ and PVP being relatively large, we hypothesized that small ligands might not show cooperative adsorption because they are less likely to feel each other on surfaces. We chose halides (I⁻ and Br⁻) and a small thiol β-mercaptoethanol (BME) as representatives for their wide applications in nanoparticle chemistry (e.g., in surface functionalization and self-assembled monolayers)[3,23]. Indeed, COM-PEITS titrations on single 5-nm Au nanoparticles give $h = 1$ for all three, indicating no adsorption cooperativity (Fig. 1d and Supplementary Fig. 12g–i), besides their adsorption affinities (Fig. 1h). We note $K_{Br^-}$ is only ~0.2% of $K_{CTAB}$, confirming that

CTAB adsorption is dominated by its CTA⁺ cation. The counter-cation K⁺ for I⁻ and Br⁻ has insignificant adsorption (Supplementary Information section 2). The noncooperative adsorption of BME is consistent with bulk measurements of 4-aminothiolphenol adsorption on pseudospherical Au nanoparticles[17]; notably, when these particles were roughened to create highly concave surfaces, positive cooperativity of 4-aminothiolphenol adsorption ($h$ ~2) emerged concurrently with >10¹ times weaker affinity[17], consistent with our observation that weaker adsorbate–surface interactions can lead to stronger adsorbate–adsorbate interactions.

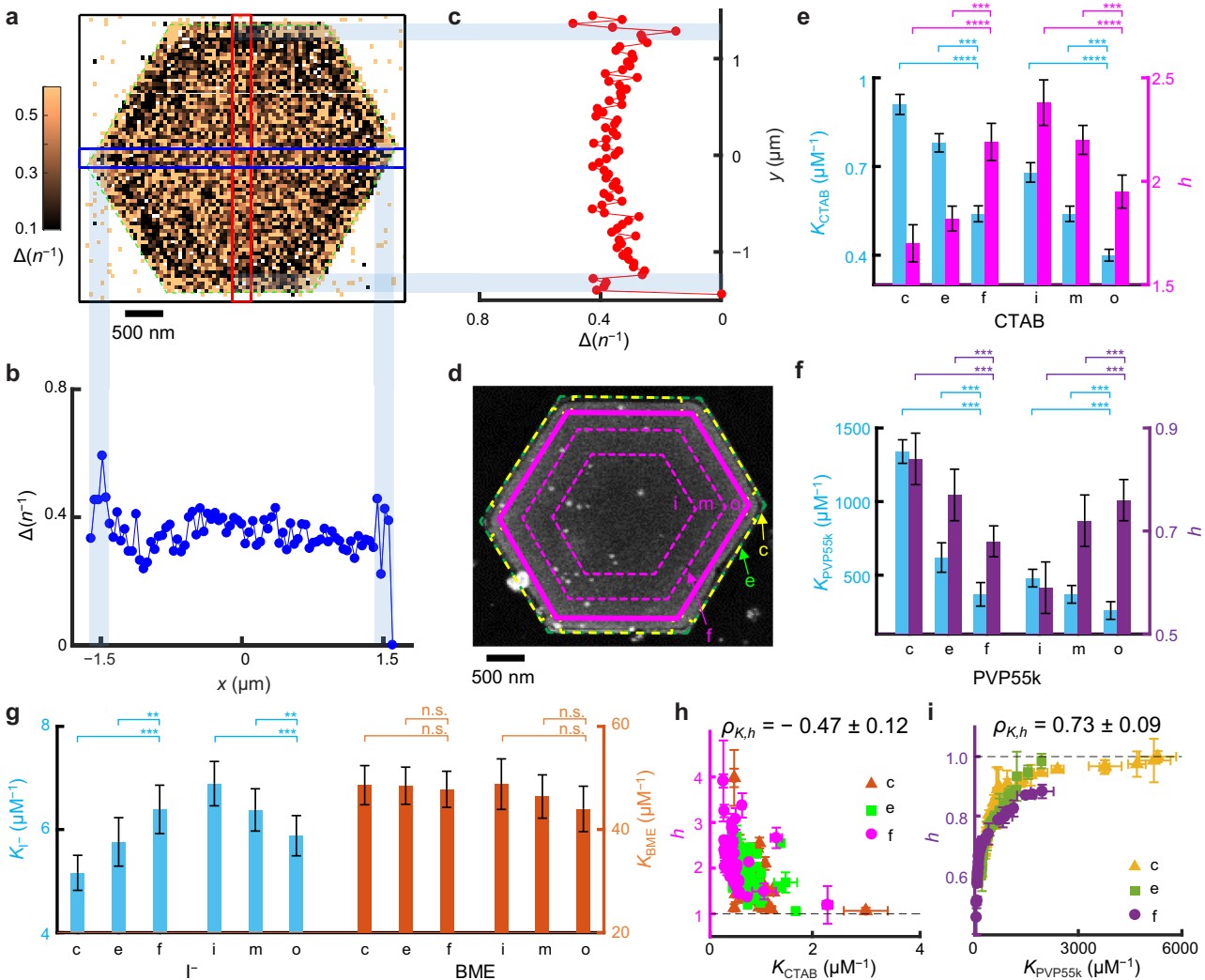

**Fig. 2 Sub-particle variations of adsorption affinity and cooperativity on Au nanoplates. a** Representative COMPEITS image ($40^2$ nm²/pixel) of a Au nanoplate for CTAB adsorption calculated between [CTAB] = 0.5 and 0 μM. $n$: number of fluorogenic auxiliary reaction products detected over 45 min. $\Delta(n^{-1}) \propto \Delta(v_R^{-1}) \propto K_L^h$ based on Eq. (1) and Eq. S13. White/null pixels: occasional negative values or infinities from 1/0 calculations. **b, c** 1D projections of blue/red-boxed regions. **d** Corresponding SEM and scheme of segmentation (Supplementary Fig. 13a and Supplementary Fig. 7). Green dashed line: fitted outer contour of the mesoporous silica (~40 nm thick; Supplementary Fig. 1a–f) coated nanoplate; magenta solid line: boundary of the flat facet (f) region; and the space in-between are divided into corner (c) and edge (e) regions. The facet region is further divided into three equal-area, inner (**i**), middle (**m**), and outer (**o**) radial segments, separated by magenta dashed lines. **e–g** Facet and sub-facet differences in adsorption affinity ($K$) and cooperativity ($h$) of CTAB (**e**), PVP55k (**f**), I⁻ and BME (**g**, no cooperativity) on 55, 40, 36, and 40 nanoplates, respectively. **h, i** Sub-particle $h$ vs. $K$ correlation for CTAB (**h**) and PVP55k (**i**). Each nanoplate provides one point per c, e, or f region. *$p < 0.05$; **$p < 0.01$; ***$p < 0.001$; ****$p < 0.0001$; n.s. nonsignificant ($p > 0.05$); paired Student's $t$ test. Error bars are s.e.m. in **e–g**, s.d. in **h**, **i**.

## Sub-particle/facet variations of adsorption affinity and cooperativity.

The tens-of-nanometer spatial resolution of COMPEITS enabled examining sub-particle-level ligand adsorption behaviors on larger, anisotropically shaped particles like triangular/hexagonal Au nanoplates, which are inaccessible to bulk measurements. These nanoplates are micrometers wide and ~14 nm thick, with {111} flat facets and {110} side facets (Supplementary Fig. 2a)[27], offering an effective platform for studying facet-dependent molecular adsorption behaviors. We further coated them with mesoporous silica (Fig. 2d and Supplementary Fig. 1a–f) to stabilize them upon removing their capping ligands that remained from the synthesis, and to prepare them for ligand adsorption studies (Supplementary Information section 1.2.3).

We first mapped CTAB adsorption as a representative for positive cooperativity. The COMPEITS image, which is the inverse difference $\Delta(n^{-1})$ between the super-resolution images of

the fluorogenic auxiliary reaction in the absence and presence of CTAB (Eq. S13), directly resolves adsorption differences at the corner (c), edge (e), and flat-facet (f) regions on any nanoplate in correlation with its SEM image (Fig. 2a–d, and Supplementary Information section 1.8.4). Titrations of $v_R$-vs-[CTAB] gave the respective adsorption equilibrium constant $K$ and Hill coefficient $h$ of each region on each of 55 nanoplates. They show clear site-dependent CTAB adsorption affinity, with $K_c > K_e > K_f$ (Fig. 2e), which we attribute to that CTAB binds on Au{110} more strongly than on Au{111} (i.e., $K_{CTAB}^{\{110\}} > K_{CTAB}^{\{111\}}$) and that corner regions have a higher {110} portion than edge regions (see Supplementary Information section 5.5 on discussions of under-coordinated atom contributions). This trend is consistent with that EELS imaging detected more adsorbed CTAB on Au{110} than Au {111}[20]—it is worth noting that the adsorbed amount does not necessarily report affinity (see later) and EELS is an ex situ

measurement. More important, the cooperativity follows $1 < h_c < h_e < h_f$ (Fig. 2e), reflecting a facet-dependent cooperativity (i.e., $1 < h_{CTAB}^{\{110\}} < h_{CTAB}^{\{111\}}$), here within a single particle. Again, $h$ is anti-correlated with $K$ as earlier (Fig. 2e; same for individual nanoplates, Fig. 2h).

We note that the mesoporous silica shell here does not render the cooperativity, nor should it bias the trends of cooperativity across different regions because of the following: (1) Cooperativity is observed directly on uncoated 5-nm nanoparticles (Fig. 1). (2) All $h$ values here suggest cooperative adsorption of two CTAB molecules (the width of a stretched $CTA^+$ is below 1 nm and its length is ~2 nm[28,29]), which are smaller than the pores of the shell (~3.5 nm on average[30]). (3) The anti-correlation between $K$ and $h$ for CTAB adsorption is persistently observed in the absence (i.e., 5-nm nanoparticles) or presence (i.e., nanoplates here and nanorods later) of the shell. (4) The Au{111} facet shows stronger cooperativity than the Au{110} facet, regardless of whether the {111} facet is located dominantly at low curvature regions (i.e., the top flat facet of nanoplates) or high curvature regions (i.e., at the tips of nanorods; later) of the particle. (See more discussion on the insignificant role of the shell in cooperativity in Supplementary Information section 1.2.3).

The nanoplate's large flat {111} facet further allowed for dissecting it into three radial segments: inner (i), middle (m), and outer (o) (Fig. 2d). We previously established that on the flat {111} facet, the structural defects decrease in density from the center toward the periphery because of their seeded growth mechanism[31]. The determined CTAB affinity follows this trend of defect densities (Fig. 2e), attributable to CTAB's preferred binding to structural defects. Interestingly, $h_i > h_m > h_o$, an observation of sub-facet cooperativity differences, is positively correlated with the affinity trend of $K_i > K_m > K_o$, opposite to that among the corner, edge, and facet regions (Fig. 2e). Here, we attribute that the inner segment has a higher defect density (the underlying reason of larger $K$), and consequently a higher density and shorter inter-distances of adsorbed CTAB; this shorter inter-distances should render more adsorbate–adsorbate interactions, dominating over the adsorbate–surface interactions within the same facet, and thus overall stronger cooperativity.

We then studied PVP55k on nanoplates as a representative for negative cooperativity. At the sub-particle level, $K_c > K_e > K_f$ (Fig. 2f), indicating PVP's preferred binding to Au{110} than Au{111} (i.e., $K_{PVP}^{\{110\}} > K_{PVP}^{\{111\}}$), similar to CTAB; this trend is also consistent with that ex situ NanoSIMS imaging detected more adsorbed PVP on {110}[19], although the adsorbed amount does not reliably reflect affinity. Concurrently, $1 > h_c > h_e > h_f$ (Fig. 2f), showing less cooperativity with stronger binding facets as earlier (also Fig. 2i), i.e., $1 > h_{PVP}^{\{110\}} > h_{PVP}^{\{111\}}$. At the sub-facet level within {111}, PVP displays the same variations in affinity and cooperativity as CTAB, with $K_i > K_m > K_o$ and $h_i < h_m < h_o < 1$ (Fig. 2f). PVP10k shows identical trends (Supplementary Fig. 15g).

As comparisons, we studied $I^-$, $Br^-$, and BME that show no adsorption cooperativity. For $I^-$, $K_c < K_e < K_f$ (Fig. 2g), opposite to CTAB and PVP, likely because iodide prefers the coordination geometry on Au{111} than on Au{110} (i.e., $K_{I^-}^{\{110\}} < K_{I^-}^{\{111\}}$). This result is supported by calculations[32] and agrees with NanoSIMS measurements[19]. Its sub-facet variation of $K$ within {111} is like CTAB and PVP (Fig. 2g). $Br^-$ shows the same trends as $I^-$ (Supplementary Fig. 15h). In contrast, BME shows non-differentiating adsorption: $K_c = K_e = K_f$, and $K_i = K_m = K_o$ (Fig. 2g). This structural insensitivity of BME adsorption likely stems in part from its exceptionally strong affinity toward Au and underlies the wide-spread usage of thiols in stabilizing pseudospherical nanoparticles[33] and in surface-functionalizing particles non-discriminatingly[30].

Compared with nanoplates, penta-twinned Au nanorods also feature an anisotropic morphology but possess complementary facets, with the larger side facets and two tips covered by {110} and {111} facets, respectively[34] (Supplementary Information section 1.3 and 1.4 and Supplementary Fig. 2b). We studied penta-twinned Au nanorods of ~1000 nm in length and ~35 nm in diameter[35], also coated with mesoporous silica (~40 nm thick; Supplementary Fig. 1g–k), for adsorbing molecules with distinctive cooperativities. COMPEITS imaging in correlation with SEM directly resolves differences in adsorption behavior at the two tip regions (T) vs. the side facets (S) (Fig. 3a–c and Supplementary Fig. 7c). Titration analysis gave $K$ and $h$ values at respective regions of each nanorod: $K_T < K_S$ for CTAB and PVP (Fig. 3d, e) and $K_T > K_S$ for $I^-$ and $Br^-$ (Fig. 3f), consistent with and further corroborating our earlier attribution that $K^{\{110\}} > K^{\{111\}}$ for CTAB/PVP and opposite for $I^-/Br^-$[19,20]. Consistently, for CTAB and PVP, stronger binding leads to weaker cooperativity between different facets both on average (Fig. 3d, e) and at the single-particle level (Fig. 3g, h).

To study the sub-facet dependence on single nanorods, we further dissected the {110} sides into inner (I), middle (M), and outer (O) regions from the center toward the tips (Fig. 3b and Supplementary Fig. 7c). We previously established that on the side facets of these nanorods, surface defect densities decrease from the center toward the two tips resulting from their seeded-growth synthesis[36]. For CTAB, PVP, and $I^-$ and $Br^-$, we observed $K_I > K_M > K_O$ (Fig. 3d–f), likely dictated by the decreasing defect density away from the center. Within the {110} side facets, $h$ becomes closer to 1 and thus less cooperative with increasing $K$ (Fig. 3d, e), similar to the trend observed within the {111} flat facets of nanoplates. These consistent results further corroborate our attribution that the defect density gradient underlies the differences in affinity and cooperativity within the same facet on a single particle, regardless of the underlying facet being {111} for nanoplate's flat facets or {110} for nanorod's side facets.

**Crossover behavior of ligand adsorption between different facets.** Provided the adsorption affinity and cooperativity, one can determine the density of adsorbed ligands ($\rho$) on any facet at any [L]:

$$\rho = \rho_{max} \frac{(K_L[L])^h}{1 + (K_L[L])^h} \qquad (2)$$

where $\rho_{max}$ is the ligand's maximal packing density on the facet. $\rho$ is a measure of a facet's stabilization by ligand adsorption and of its accessibility to other reagents. Larger $\rho$ is often presumed to reflect stronger adsorption affinity on a facet[5,19,20]. However, even for Langmuir adsorption (i.e., non-cooperative with $h = 1$), the density of adsorbed ligands ($\rho^{strong}$) on a stronger-binding facet ($K^{strong}$) vs. that ($\rho^{weak}$) on a weaker-binding facet ($K^{weak}$) can have complex behaviors. Only when $\rho_{max}^{strong} \geq \rho_{max}^{weak}$, the presumption is valid and $\rho^{strong} \geq \rho^{weak}$ at any [L] (Fig. 4a, b). But, when $\rho_{max}^{strong} < \rho_{max}^{weak}$, the two $\rho$-vs-[L] curves cross over at a critical ligand concentration ($c_x$) (Supplementary Eq. S23), defining two distinct regimes (Fig. 4c): at [L] < $c_x$, $\rho^{strong} > \rho^{weak}$ because the adsorption affinity plays a dominant role; at [L] > $c_x$, $\rho^{strong} < \rho^{weak}$ because $\rho_{max}$ limits the amount of adsorbates. This crossover behavior stems from the interplay between multiple factors (e.g., adsorption affinity, density, and/or cooperativity) that determine the overall molecular adsorption properties. This behavior also predicts that the relative stabilization and

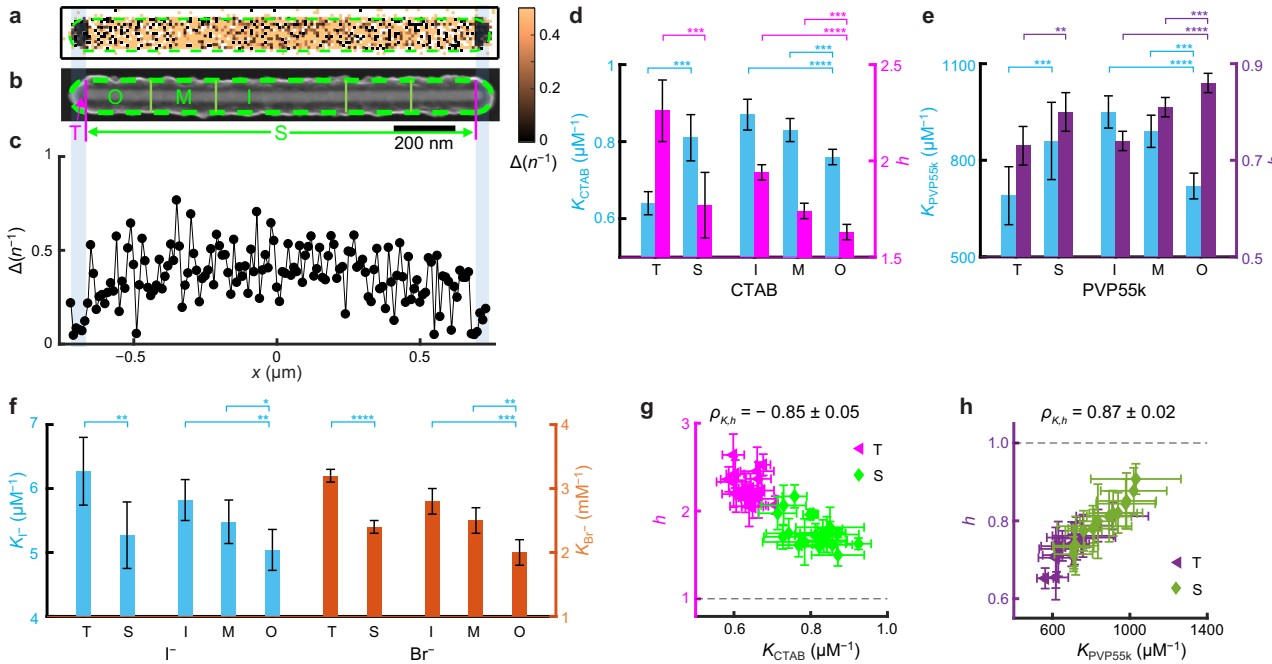

**Fig. 3 Sub-particle variations of adsorption affinity and cooperativity on Au nanorods. a–c** COMPEITS (**a**) and SEM (**b**) images as in Fig. 2a–d, but for a Au nanorod ($10^2$ nm²/pixel); the 1D projection (**c**) is for the entire nanorod. **d–f** Facet and sub-facet differences in adsorption affinity ($K$) and cooperativity ($h$) of CTAB (**d**), PVP55k (**e**), and I⁻ & Br⁻ (**f**) on 20, 15, and 21 & 44 nanorods, respectively. **g, h** Sub-particle $h$ vs. $K$ correlation for CTAB (**g**) and PVP55k (**h**). Each nanorod provides one point per T or S region. *$p < 0.05$; **$p < 0.01$; ***$p < 0.001$; ****$p < 0.0001$; n.s. nonsignificant ($p > 0.05$); paired Student's $t$ test. Error bars are s.e.m. in **d–f**, s.d. in **g, h**.

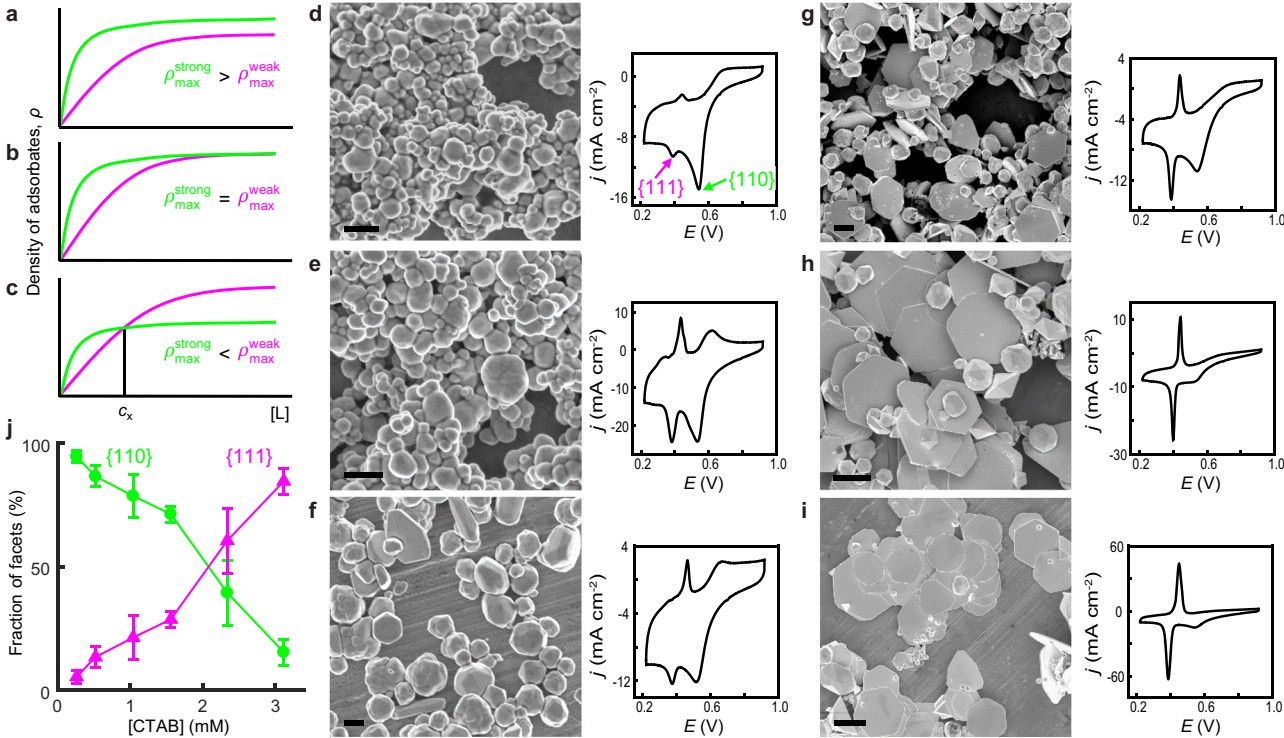

**Fig. 4 Crossover behavior of ligand adsorption on different facets and application to facet-tuned Au nanoparticle synthesis. a–c** Three scenarios for adsorbed ligand density $\rho$ as a function of [L] between a strong (green) and a weak (magenta) adsorbing facet based on Eq. (2). **d–i** Left: SEM images of Au nanoparticles synthesized at increasing [CTAB]: 0.26 (**d**), 0.52 (**e**), 1.04 (**f**), 1.56 (**g**), 2.34 (**h**), and 3.12 mM (**i**). Right: corresponding CV of underpotential Pb deposition (Supplementary Information section 1.4); *j*: current density. The Pb deposition peaks on Au{111} and Au{110} are denoted in **d**. All potentials are relative to RHE. Scale bars: 500 nm in **d–f**; 2 μm in **g–i**. **j** Fractions of Au{110} and Au{111} facets vs. [CTAB] derived from CVs in **d–i**.

accessibility of various facets on a nanoparticle can markedly change depending on the ligand concentration, which has many possible applications in materials design, surface modification, and catalysis (Supplementary Information section 6.6).

One possible application of this crossover behavior would be facet control in colloidal nanoparticle synthesis—one could shift the dominance between different facets by simply tuning the concentration of a single ligand. We therefore proceeded to synthesize colloidal Au nanoparticles in the presence of increasing [CTAB], via reduction of $HAuCl_4$ by ascorbic acid (AA) in aqueous solutions in one pot, where [$HAuCl_4$] and [AA] were fixed and their adsorptions on the nanoparticle are insignificant comparatively (Supplementary Information section 1.9 and 6.1–6.2). We chose CTAB because it is redox-inert and a facet-directing stabilizer for Au nanoparticle synthesis[4], whereas PVP is oxidizable[24] and halides are ineffective stabilizers[3]. As $K_{CTAB}^{\{110\}} > K_{CTAB}^{\{111\}}$ and if the crossover behavior applies here (i.e., $\rho_{max}^{\{110\}} < \rho_{max}^{\{111\}}$), one predicts that Au{110} would be the prevalent facet at low [CTAB] whereas Au{111} would dominate at high [CTAB] (Fig. 4c). Excitingly, when [CTAB] increases from ~0.2 to ~3 mM (below CTAB's critical micelle concentration at the synthesis temperature of 85 °C; Supplementary Information section 6.3)[21,37], the synthesized nanoparticles indeed show a shape progression: initially irregular in shape, then a mixture of irregular-shaped nanoparticles and hexagonal nanoplates, and eventually high-purity nanoplates (Fig. 4d–i). The prevalence of {111}-dominated nanoplates at high [CTAB] directly supports the existence of crossover behavior of CTAB adsorption on Au nanoparticle surfaces.

To quantify facet distributions on these samples, we performed electrochemical Pb underpotential deposition, whose deposition potentials on Au{111} and Au{110} are resolvable in cyclic voltammetry (CV) at ~0.38 and ~0.53 V vs. RHE, respectively[38]. Indeed, the {110} deposition peak in CV gradually diminishes while the {111} peak grows across the set of Au nanoparticles synthesized at increasing [CTAB] (Fig. 4d–i). Consistently, the corresponding two facet-fraction-vs.-[CTAB] curves intersect at ~2 mM CTAB (Fig. 4j), below which {110} dominates and above which {111} dominates. This intersection verifies CTAB's crossover adsorption behavior and its ability in controlling facet distributions.

## Discussion

Understanding molecular adsorption quantitatively, especially at the nanoscale, is of great significance owing to its decisive role in determining the efficacy of processes such as in materials control, catalysis, and separation. The high-resolution, quantitative knowledge from nanoscale imaging offers not only molecular insights (e.g., adsorption cooperativity and crossover) but also control parameters for facet-controlled syntheses of colloidal nanoparticles. One could also envision a myriad of other potential applications, e.g., in nanoparticle surface carving via selective etching, ligand-induced galvanic replacement in generating hollow nanostructures, facet-selective deposition on solid particles, tunable surface functionalization, catalysis selectivity control, and catalyst poisoning mitigation (see discussions in Supplementary Information section 6.6). For all of these, we believe that the approaches untaken and the molecular insights gained here should open windows into uncharted scientific territories.

## Methods

**Bulk measurements of resazurin reduction and ligand adsorption competition on Au nanoparticles.** Bulk experiments were performed based on the titration of catalytic activities of Au nanoparticles in the absence (and then presence) of ligands. Specifically, pseudo-spherical colloidal Au nanoparticles, 5 nm in diameter nominally (Ted Pella 15702-20), were used to catalyze the reduction of resazurin

(R) to resorufin by $NH_2OH$, which was provided in excess (Supplementary Fig. 3a), and monitored by UV–Vis absorption spectrometry. Typically, 100 μL of the 5-nm Au suspension (0.010–0.10 nM based on the number of particles) was added to a premixed 7 mM phosphate butter solution (pH = 7.4) containing different amounts of R (1.0–10 μM) and an excess amount of $NH_2OH$ (1 mM). The reaction mixture turned gradually from blue to red, and the absorption peak at 602 nm (R) decreased while the absorption peak at 572 nm (resorufin) increased (Supplementary Fig. 3b). For the COMPEITS experiments, the reaction conditions were kept the same except that [R] was fixed while ligands with increasing concentrations were added into the reaction mixtures. The ligand concentration ranges from nM to mM depending on the ligand adsorption affinity. Typically, at least 3 concentrations of R and 4 concentrations of ligands are included for the extraction of ligand adsorption affinity and cooperativity.

**Syntheses and characterization of mesoporous-silica-coated Au nanoplates and nanorods.** Au nanoplates were synthesized following a procedure modified from previous reports[31,39] (Supplementary Information section 1.2.1). Penta-twinned Au nanorods were synthesized in a three-step seed-mediated growth method following the literature[35] (Supplementary Information section 1.3.1). The as-synthesized Au nanoplates and nanorods were coated with mesoporous silica in three major steps as previously reported[30,31,36,40,41] (Supplementary Information section 1.2.2). The morphology and shape yield of the $mSiO_2$-coated nanoplates and nanorods were examined by both TEM and SEM.

**Electrochemical underpotential deposition of Pb on Au nanoparticles for facet determination.** The underpotential deposition (UPD) of Pb on Au nanoparticles were carried out in a three-electrode cell using an electrochemical workstation (CHI 1200a potentiostat) following literature[38,42]. A Ag/AgCl electrode and a Pt wire served as the reference and counter electrodes, respectively. The as-synthesized Au nanoparticles with various morphologies were drop-casted on a glassy-carbon electrode and used as the working electrode. Before the cyclic voltammetry (CV) measurements, each sample was first subjected to potential cycling from −0.2 to −0.7 V (vs. Ag/AgCl) at a scan rate of 100 mV s$^{-1}$ until the voltammograms were stable. Afterward, three cycles of CV curves were recorded in potentials ranging from −0.2 to −0.7 V at a scan rate of 50 mV s$^{-1}$ in a mixture containing 0.1 M NaOH and 1 mM Pb(NO$_3$)$_2$ at ambient conditions. The third cycle of the CV curves was used to calculate the areas of deposition peaks of various facets for quantitative comparisons[38,42]. The charge values associated with the deposition of a monolayer of Pb are 444 and 330 μC cm$^{-2}$ for Au{111} and Au{110} facets, respectively. The potentials involved in this work were derived in reference to $E$(Ag/AgCl) and presented as in reference to $E$(RHE) in the main text, where RHE denotes the reversible hydrogen electrode, according to the formula $E$(RHE) = $E$(Ag/AgCl) + 0.1976 + 0.05916 × pH. The deposition and stripping of Pb on Au{111} and Au{110} show as two pairs of distinctive reduction–oxidation peaks at 0.38 and 0.45 V and 0.53 and 0.67 V, respectively, as reported by Wuttig et al.[38].

**COMPEITS imaging experiments and data analysis.** All single-molecule fluorescence microscopy experiments for COMPEITS imaging were carried out on a home-built prism-type wide-field total internal reflection fluorescence (TIRF) microscope (Supplementary Information section 1.7).

Information of single-molecule catalysis was extracted using a home-written MATLAB program from the fluorescence images in the movies, 'subtraction iQPALM' (image-based quantitative photo-activated localization microscopy, see Supplementary Software 1), which was expanded from iqPALM[43,44] and whose major steps of data analysis were described and employed in our previous work[11]. See more details in Supplementary Information section 1.8.

**Facet-controlled synthesis of colloidal Au nanoparticles using CTAB as the ligand.** Au nanoparticles were synthesized via reduction of $HAuCl_4$ by AA in the presence of various [CTAB] in aqueous solution, modified from an earlier protocol[45]. In a typical synthesis, 42 μL of 10 mM aqueous $HAuCl_4$ solution was added into 716 μL of CTAB aqueous solution in a plastic tube. The mixture was shaken by hand for a few times before leaving undisturbed. After 15 min, 42 μL of 10 mM aqueous AA solution was added into the tube in one shot, followed by rapid inversion for 10 times. The concentration of CTAB in the final mixture ranges from 0.26 to 3.12 mM. The tube was then placed in a water bath at 85 °C for 1 h. The products were collected via centrifugation, washed with a mixture of DI water and ethanol (1:1 v/v) followed by centrifugation at 100 g for three times, and then re-dispersed in DI water for further characterizations.

## Data and materials availability

All data are available in the main text or the supplementary materials. Raw data supporting the findings of this study and home-synthesized Au particles are available upon reasonable request.

## Code availability

MATLAB codes are included in the Supplementary Software 1.

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

## Acknowledgements

We thank M. Hesari for helping with imaging and C. Liu for discussions. The research is primarily supported by the Army Research Office (grant no. W911NF-17-1-0590). The study on halides is supported by the Center for Alkaline-based Energy Solutions (CABES), an Energy Frontier Research Center of U.S. Department of Energy under Award DE-SC0019445. The research used Cornell Center for Materials Research Shared Facilities supported by NSF (grant no. DMR-1719875). R.Y. thanks Cornell Presidential Postdoctoral Fellowship for support.

## Author contributions

R.Y. designed imaging experiments, synthesized particles, performed bulk titration and single-molecule imaging experiments, derived kinetic models, wrote computer codes, and analyzed results. M.Z. designed and performed facet-controlled particle synthesis, performed structural and electrochemical characterizations, and analyzed results. X.M. wrote computer codes, discussed experiments and results, and contributed to manuscript revisions. Z.W. contributed to imaging experiments. D.A.G. and H.P. contributed to particle synthesis and bulk kinetic measurements. Z.Z. contributed to data visualization. R.Y., M.Z., and P.C. discussed results and wrote the manuscript. P.C. directed research.

## Competing interests

The authors declare no competing interests.
