## [Peer Review File · Nature Communications]

REVIEWER COMMENTS

Reviewer #1 (Remarks to the Author):

Referee report on the paper 'Nanoscale cooperative adsorption for materials control' by Rong Ye et al. In this paper, the authors use their recently developed COMPEITS super resolution technique (based on competitive adsorption) to study adsorption on metal nanoparticles. They deliver evidence of cooperative adsorption, both positive and negative, on individual particles and even facets within individual particles. The authors also show cross-over behavior of ligand adsorption on different facets and use the concept to tune facet exposure in NPs. The paper has an extensive SI section that contains a lot of useful information. The results reported are interesting to the general public of Nature Communications, the paper is well-written, and the topic is timely. I can recommend publication of the paper if following questions are addressed:

A good testcase for the concepts put forward here would be proving these concepts on the different nanoparticle allotropes that exist for gold, silver, ruthenium... Is this something the authors have data on? It would strengthen the claims made in the paper.

The resolution achievable with COMPEITS doesn't allow to resolve the first test sample (5 nm gold particles) and hence the reported values (hill coefficients etc) are particle averaged (averaged over several tens of particles). How large is the spreading of the data and to what can the differences be attributed?

I am not convinced about the reasoning that the porous silica shell is not influencing the cooperativity: saying that coefficients are similar than those of the 5 nm particles can be hardly called direct evidence. As the silica shell and the ligands are present next to each other (and competing for the same space), one can not simply ignore its presence. Have the authors additional evidence that the porous silica has NO influence at all as claimed?

As for the nanoplates, how was subdividing the particle in sections decided, it seems a little arbitrary now. The magenta zones in the fig2, a, IV hard to seen in the figure by the way.

Would this concept also work on mixed metal NPs, e.g. silver/gold nanowires? Can the differences in adsorption be accounted for?

Reviewer #2 (Remarks to the Author):

Peng Chen and co-workers have used their recently developed COMPEITS method to assess the process of adsorption from a fundamental level at the nanoscale using gold nanoparticles. There are three important areas of application in which the method is explored, and cooperativity has been assessed. I specifically like the implementation on the facet-controlled synthesis of colloidal metal nanoparticles. This aspect is certainly very innovative. I have one comment to make on the article; I believe it would benefit by trying to get the mathematical analysis of the data on correlation better explained and illustrated with schematics/figures. The work is at several places not so well-understandable and also the effects are not that dominant/evident for the reader. In this respect, some of the figures are very

crowded with many data overlaying each other, not making it always easy to understand what the authors wishes to say. Hence, some rewrite/restructuring of the text and data are needed. Nevertheless, the article is very innovative and also brings new insights in an important topic for a wide scientific audience, not per se limited to chemistry.

Reviewer #3 (Remarks to the Author):

In the manuscript "Nanoscale cooperative adsorption for materials control", the authors describe the application and extension of COMPEITS (a methodology recently reported by the same group) to study ligand adsorption onto a variety of Au nanoparticles, focusing on the kinetics of different ligands and nanoparticle morphologies. The authors also produced a study of nanoparticle synthesis where a selected ligand concentration was adjusted to favor growth on different facets, resulting in a degree of control over morphology. The work is clearly presented and appears scientifically sound by the judgment of this reviewer. Furthermore, it provides interesting insights into the surface interactions of nanoparticles, and is a clear advancement of the COMPEITS approach.

While the COMPEITS study in the first part of the manuscript is comprehensively quantitative, the synthesis study in the second part is not, and seems an after-thought and incomplete. The role of ligand affinity for specific facet types is frequently used to control nanoparticle shape during synthesis. In this work, a "cross-over model" is used to explain the behavior qualitatively, but does not predict. There does not appear to an attempt to obtain ρ_{\max} from the single-particle results nor estimate the cross-over point, only implying that there should be one considering the relative magnitudes of adsorption equilibrium constants on different facet regions. This does not appear to be a substantial enough point. In particular, the environment of a synthesis and the environment of the single-particle measurements are different enough that the ligand behavior may not be the same under both conditions.

However, I do recommend this manuscript be published in Nature Communications, after revision addressing the concerns listed below and after taking into consideration the appropriateness of the synthesis results.

1. 5nm spheres have high surface curvature. Does the rate model (Eqn. 1) need to be modified for planar versus curved surfaces?
2. On a planar surface, the cooperativity for long chains would be expected to be lower/negative, but for spheroids, is that the case?
3. Is there an underlying assumption that resazurin adsorption is independent of the facet type? A related question: do different facets have different optimal resazurin concentrations for COMPEITS measurements?
4. The correlation between h and K is weaker for $h > 1$ and stronger for $h < 1$ (Figs. 1i & 1j). Is there an explanation for why? Line 90-92 points out the correlations, but the correlations are not on equal footing.

5. The error bars in Figs. 1f & 1g are substantially narrower than the distribution of measurements in Figs. 1i & 1j (also in Fig. 2). It is noted that one is s.e.m. and the other is std. However, the difference is confusing and misleading.
6. How narrow is the pH range where resorufin is brightly fluorescent? Could the pH conformation dependence of PVP be explored by going outside of the optimal range for resorufin?
7. How is the optimal resazurin concentration determined? Does it change for particle geometry or ligand selection?
8. Were the commercial 5 nm Au NPs also coated in silica? Were the smaller particles also imaged with SEM for to check for clustering?
9. The arguments about cooperativity with the silica in lines 161-167: where is the evidence is does not alter behavior? Yes, there is evidence that cooperativity occurs, but why can it not be different than for the small particles (with high surface curvature)? Is K_L at least affect by the reduced surface area?
10. It would be useful to see the raw super-resolution images for individual nanoparticles. The COMPEITS images appear to be binned, but high-resolution images of both $[L] = 0$ and $[L] > 0$ would be informative (using marker transparency to indicate localization density). If claiming 10 nm localization resolution, it would be good to see un-binned results. Also, for fluorescence that appears in multiple frames, it would be interesting to see their distribution of localizations over time.
11. In flow cells, there can be regions of low/no flow due to channeling. If titrating multiple ligand concentrations during a measurement series, how can one be sure there is nothing left from the previous titration? Are the flow cells evacuated between concentrations? There could be a distribution of concentrations across the FOV. Although the individual rate curves suggest this is not an issue due to their high-fidelity fits.
12. The frequent use of hyperbolic terms such as "unprecedented" (P5, P9) and "first-of-its-kind" (abs., P3, P8, P16) are, at worst, untrue exaggerations and, at the least, simply unnecessary and distracting. Please respect the intelligence of your reader to understand the significance of your results. It is a misconception that such phrases actually improve your rate of manuscript acceptance.
13. Is fluorescence from an edge or corner region at least partially physically obstructed? With the high-NA object, light will be collected given the NP sizes, but is there evidence of lower fluorescence intensities near corners and edges?
14. Within a geometric feature (face, edge), is there is mix of facet types that can be predicted or otherwise determined? For example, on a face, Au NPs are not necessarily atomically smooth. Do steps generate mixtures of facet types and can the ratio of the facet types be estimated?
15. The clustering of data in Fig. 3a well-separates the different features. Does Fig. 2h have similar quality clusters if similarly scaled and outliers/tails of the distribution are ignored?
16. Do the geometry differences from synthesis with varying [CTAB] stem from initial seeding of the growth or is it a result of ligand adsorption throughout the synthesis time? Could a synthesis seeded with nanorods dominated by {110} be converted into nanoplates dominated by {111}? Insufficiently exploring the synthesis characteristics is another reason why this part of the study appears an after-thought.
17. It is unclear if "defect sites" are necessary for COMPEITS (or adhesion in general) or simply provide regions of enhanced activity. The authors casually mention defect density in explanations, but do not define what a defect site is. Are defects regions where the crystalline structure is abruptly

inhomogeneous? In that case, does that inaccurately probe the surface in those regions, making this technique sensitive to heterogeneities and not the facet types?

18. From previous works (Ref. 30), the defect density variations across a nanoparticle seems to be around 50%. Can the authors speak to expected defect density changes from i/m/o? Is there a length scale from the center of a nanoparticle where the inner region is constant and only deviates within a characteristic distance from an edge?

19. Although the COMPEITS method does not necessarily require precise super-resolution localization methods since the detection rate is the important component, is it surprising the decisions the authors made for their localization analysis. The background subtraction method is ad hoc (rather than including background emission rates in the fitting term “b”) and inaccurately represents noise in the images, particularly in low-intensity pixels. However, because the authors go through the effort to claim high localization precision in Section S1.8, I must object to their approach. It appears in this section that authors incorrectly apply Eqn. S18 in Eqn. S19. The former produces an estimate of the photon flux impinging on the camera. However, the latter, used to determine localization precision, is based on the number of DETECTED photons. Photons that are not detected cannot improve localization precision and applying these equations as they appear artificially improves the author’s precision estimate.

Furthermore, the citations here (Refs. 35 & 54) are references to the author’s previous works. However, the derivation of the precision equation was presented in other works and should be cited correctly (i.e., “Precise Nanometer Localization Analysis for Individual Fluorescent Probes” by Thompson, et al., 2002). It is frustrating that the authors seem to have chosen to cite their own work more prolifically than foundational papers.

20. Despite my objections to the treatment of the background emission, it is also odd that the authors take care to accommodate the sub-pixel shifts of the background images, but still use a Gaussian PSF instead of a pixel-integrated PSF. Why? This has ramifications for the estimated intensities, localization, and the precision analysis. It can also lead a optimization algorithm to erroneous results (such as the PSF width) even for a true detection event.

21. The SI indicates how non-events and double-detection events are treated in analysis. It would be useful to see the spatial distribution of the rejection rates to see if there is a bias among c/e/f or i/m/o regions of a nanoparticle. Particularly since localization in the edge regions are susceptible to the noise concerns from the background subtraction described above, the authors should demonstrate the rejection rates of detected events in these regions are unaffected by geometry.

Reviewer #1 (Remarks to the Author):

Referee report on the paper 'Nanoscale cooperative adsorption for materials control' by Rong Ye et al.

In this paper, the authors use their recently developed COMPEITS super resolution technique (based on competitive adsorption) to study adsorption on metal nanoparticles. They deliver evidence of cooperative adsorption, both positive and negative, on individual particles and even facets within individual particles. The authors also show cross-over behavior of ligand adsorption on different facets and use the concept to tune facet exposure in NPs. The paper has an extensive SI section that contains a lot of useful information. The results reported are interesting to the general public of Nature Communications, the paper is well-written, and the topic is timely. I can recommend publication of the paper if following questions are addressed:

[Reply 1] Thank you very much for the appreciation of our work. Please also see below our point-by-point responses to your other comments, which we have considered carefully and revised the manuscript accordingly. The major revisions are in red fonts in the marked copies of the main text and supplementary information.

A good testcase for the concepts put forward here would be proving these concepts on the different nanoparticle allotropes that exist for gold, silver, ruthenium... Is this something the authors have data on? It would strengthen the claims made in the paper.

[Reply 2] Regarding “the concepts”, are you referring to using COMPEITS to map molecular adsorption at the nanoscale, or the cross-over behavior of molecular adsorption between different facets? For the former, we currently only have data on Au particles, in addition to that on BiVO₄ particles in our previous work (*Nat. Chem.* **2019**, *11*, 687). We do expect COMPEITS to be applicable to other metal particles and plan to explore them in the future. The general applicability of COMPEITS is also discussed in our previous work (*Nat. Chem.* **2019**, *11*, 687 and the supporting information).

For the latter, we currently do not have data other than Au particles. As we discussed in Supplementary Information section 6.5, the cross-over behavior can exist only if $\rho_{\max}^{\text{strong}} < \rho_{\max}^{\text{weak}}$, which may not always be applicable. In light of your comment, we expanded the discussion in this Supplementary Information section, now reads: “Note for c_x to have a positive value, which is a prerequisite for the application of the cross-over concept for shape-controlled synthesis of other metals/materials, $\rho_{\max}^{\text{weak}}$ has to be greater than $\rho_{\max}^{\text{strong}}$.”

The resolution achievable with COMPEITS doesn't allow to resolve the first test sample (5 nm gold particles) and hence the reported values (hill coefficients etc) are particle averaged (averaged over several tens of particles). How large is the spreading of the data and to what can the differences be attributed?

[Reply 3] The resolution of COMPEITS does not allow resolving *within* a single 5-nm Au nanoparticles, but does allow resolving *among* individual particles that are well-dispersed on the slide. So we do have K and h values for individual 5 nm particles. The K versus h plot for CTAB and PVP adsorption on individual 5-nm particles are shown in Fig. 1i and 1j, along with the histograms of h . These single-particle data are then averaged to obtain the particle-averaged results, e.g., in Fig. 1f-1h.

We can use the ratio of standard deviation and the mean, defined as the heterogeneity index (HI), to evaluate the spreading of data from individual particles. For CTAB adsorption on 50 nanoparticles (Fig. 1i), HI for the affinity K is $0.12/0.66 = 18\%$; and HI for the Hill coefficient h is $0.44/2.07 = 21\%$. The heterogeneity can be in part attributed to the heterogeneity of particle size: the diameter of the nominal 5-nm particles from TEM is 6.0 ± 1.6 nm (*Nat. Mater.* **2008**, *7*, 992), where the HI is 27%. We have added this discussion on heterogeneity in Supplementary Information section 3.1.

I am not convinced about the reasoning that the porous silica shell is not influencing the cooperativity: saying that coefficients are similar than those of the 5 nm particles can be hardly called direct evidence. As the silica shell and the ligands are present next to each other (and competing for the same space), one can not simply ignore its presence. Have the authors additional evidence that the porous silica has NO influence at all as claimed?

[Reply 4] There are multiple evidences supporting the minimal effect of mesoporous silica shell on cooperativity (using CTAB adsorption as an example):

- 1) The magnitude of h for CTAB adsorption on 5-nm Au nanoparticles, Au nanoplates, and Au nanorods are all roughly 2. This similarity between the naked 5-nm Au nanoparticles and mesoporous-silica-coated nanoplates and nanorods confirms the minimal effects of the silica shell on the measured cooperativity.

- 2) We persistently observed the anti-correlation between K and h for CTAB adsorption on 5-nm Au (without a silica shell) and nanoplates/nanorods (coated with mesoporous silica).
- 3) The {111} facet shows stronger cooperativity than the {110} facet, regardless of whether the {111} facet is located dominantly at low curvature regions (i.e., the top flat facet on nanoplates) or at high curvature regions (i.e., at the tips of nanorods). Therefore, the presence of the mesoporous silica shell might affect the exact values of h , but it should not alter h biasedly or change the trends across regions.

We have added a shortened discussion as above in the main text (page 6) and a more detailed one in Supplementary Information Section 1.2.3.

As for the nanoplates, how was subdividing the particle in sections decided, it seems a little arbitrary now. The magenta zones in the fig2, a, IV hard to seen in the figure by the way.

[Reply 5] We described the methods of dissecting the nanoplates in detail in Supplementary Information section 1.8.4 and in Supplementary Fig 7. Briefly, from the vertices of the gold core (not those of the outmost silica shell) toward the center of the gold core by 3ε gives the vertices the division boundary between the edges versus the flat facet (the solid magenta line in Fig. 2d), where ε is the overall error of localization – 39 nm in this work. The basic idea is that each segment is at least significantly larger than the uncertainties of our spatial resolution.

We have increased the line thickness to make it more visible in Fig. 2.

Would this concept also work on mixed metal NPs, e.g. silver/gold nanowires? Can the differences in adsorption be accounted for?

[Reply 6] Do you mean applying COMPEITS to quantify the affinity and cooperativity on alloyed Ag/Au nanowires? Yes, it should work. However, if differences in adsorption at different locations on a single nanowire are observed, there would be an additional possibility for interpretation: differences in local Ag/Au composition. Please note that in our Au nanorod synthesis, no Ag is introduced, although there are literature methods that use Ag ions in synthesizing Au nanorods, thus potentially introducing Ag dopants.

Reviewer #2 (Remarks to the Author):

Peng Chen and co-workers have used their recently developed COMPEITS method to assess the process of adsorption from a fundamental level at the nanoscale using gold nanoparticles. There are three important areas of application in which the method is explored, and cooperativity has been assessed. I specifically like the implementation on the facet-controlled synthesis of colloidal metal nanoparticles. This aspect is certainly very innovative. I have one comment to make on the article; I believe it would benefit by trying to get the mathematical analysis of the data on correlation better explained and illustrated with schematics/figures. The work is at several places not so well-understandable and also the effects are not that dominant/evident for the reader. In this respect, some of the figures are very crowded with many data overlaying each other, not making it always easy to understand what the authors wishes to say. Hence, some rewrite/restructuring of the text and data are needed. Nevertheless, the article is very innovative and also brings new insights in an important topic for a wide scientific audience, not per se limited to chemistry.

[Reply] Thank you very much for your appreciation of our work. To better explain the mathematical analysis of correlation, we added a note in Fig. 1 caption citing Supplementary Information section 4.3, in which we added a brief introduction of the Pearson's cross-correlation coefficient as follows:

“Pearson's cross-correlation coefficient $\rho(x,y)$ is a measure of the strength and direction of the linear relationship between two variables x and y . It can be calculated by the following equation:

$$\rho(x,y) = \frac{\sum_{i=1}^n (x_i - \langle x \rangle)(y_i - \langle y \rangle)}{\sqrt{\sum_{i=1}^n (x_i - \langle x \rangle)^2} \sqrt{\sum_{i=1}^n (y_i - \langle y \rangle)^2}} \quad \text{Eq. S20}$$

where n is the sample size, $\langle \rangle$ denotes averaging. Thus, ρ is essentially a normalized measurement of the covariance, and always has a value between -1 and 1 : $\rho(x,y) = 1$ implies that x and y can be perfectly described by a linear equation, with all data points lying on a line for which y increases as x increases; $\rho(x,y) = -1$ implies that all data points lie on a line for which y decreases as x increases; $\rho(x,y) = 0$ implies that there is no linear correlation between the variables.”

We have also revised some text and the figures to make them clearer, including making the lines thicker in the SEM (now panel d) in Fig. 2. The major changes are in red fonts in the marked version of the main text and the supplementary information.

Reviewer #3 (Remarks to the Author):

In the manuscript "Nanoscale cooperative adsorption for materials control", the authors describe the application and extension of COMPEITS (a methodology recently reported by the same group) to study ligand adsorption onto a variety of Au nanoparticles, focusing on the kinetics of different ligands and nanoparticle morphologies. The authors also produced a study of nanoparticle synthesis where a selected ligand concentration was adjusted to favor growth on different facets, resulting in a degree of control over morphology. The work is clearly presented and appears scientifically sound by the judgment of this reviewer. Furthermore, it provides interesting insights into the surface interactions of nanoparticles, and is a clear advancement of the COMPEITS approach.

[Reply 1] Thank you very much for the appreciation of our work. Please also see below our point-by-point responses to your other comments, which we have considered carefully and revised the manuscript accordingly. The major revisions are in red fonts in the marked copies of the main text and supplementary information.

While the COMPEITS study in the first part of the manuscript is comprehensively quantitative, the synthesis study in the second part is not, and seems an after-thought and incomplete. The role of ligand affinity for specific facet types is frequently used to control nanoparticle shape during synthesis. In this work, a "cross-over model" is used to explain the behavior qualitatively, but does not predict. There does not appear to an attempt to obtain ρ_{\max} from the single-particle results nor estimate the cross-over point, only implying that there should be one considering the relative magnitudes of adsorption equilibrium constants on different facet regions. This does not appear to be a substantial enough point. In particular, the environment of a synthesis and the environment of the single-particle measurements are different enough that the ligand behavior may not be the same under both conditions.

[Reply 2] It is true that the role of ligand affinity for specific facet types is frequently used to control nanoparticle shapes during synthesis. However, in our work, a *single* ligand concentration ([CTAB]) is used as the *sole* varying parameter for facet control and for switching the dominant facet while every other experimental condition is constant. This is in contrast to that shape-controlled syntheses in the literature explore a much larger parameter space, including the ligand concentration.

Our work on the facet-controlled Au nanoparticle synthesis was actually inspired by and rationalized on the basis of our COMPEITS study, which showed different ligand adsorption affinities on different particle facets. When we attempted to relate our affinity results with literature, we realized that past studies were all measuring relative densities of the adsorbed ligand on different facets (e.g., Main text Ref 18 via NanoSIMS and Main text Ref 19 via EELS), which depend not only on the affinity but also on the maximal ligand packing density. This further made us realize that depending on the relative affinity and relative maximal packing density, the adsorbed ligand density can show a cross-over behavior between different facets upon varying the ligand concentration. To test whether such adsorption cross-over exists, we designed the Au nanoparticle synthesis experiment, in which the ligand concentration (i.e., [CTAB]) was the only tuning parameter to switch the dominant facets. We chose CTAB as the ligand because it is redox-inert and a facet-directing stabilizer for Au nanoparticle synthesis, whereas PVP is oxidizable and halides are ineffective stabilizers.

Indeed, COMPEITS cannot measure the maximal ligand packing density (ρ_{\max}). However, if ρ_{\max} is known, the cross-over concentration can be predicted using the respective affinities (K 's) as we described in Supplementary Information section 6.5 and Eq. S23. Eq. S23 is now cross-cited in the main text on line 261.

Practically, we cannot quantitatively predict the cross-over concentration under the colloidal synthesis conditions because of differences in the specific conditions (e.g., temperature). Higher temperature was needed in Au nanoparticle synthesis to achieve reasonable reaction kinetics. But the key point here is that [CTAB] was the sole varying experimental parameter, across which the dominant facet of the Au nanoparticle can be tuned as predicted by the cross-over behavior.

However, I do recommend this manuscript be published in Nature Communications, after revision addressing the concerns listed below and after taking into consideration the appropriateness of the synthesis results.

[Reply 3] Thank you again for appreciating our work.

1. 5nm spheres have high surface curvature. Does the rate model (Eqn. 1) need to be modified for planar versus curved surfaces?

[Reply 4] The rate model in Eq. 1 is generally applicable to any surface, as derived in detail in Supplementary Information section 1.5. The differences between planar versus curved surfaces should be reflected in the K and h values, provided that the difference in surface curvature could result in discernable differences in ligand-surface interactions. Please note that the measured K and h for a 5-nm Au nanoparticle represents the average of all types of surface sites on the particle.

2. On a planar surface, the cooperativity for long chains would be expected to be lower/negative, but for spheroids, is that the case?

[Reply 5] We are not sure whether we understood this question completely. Lower cooperativity, which we assume you meant weaker cooperativity, manifests differently depending on whether the cooperativity is positive ($h > 1$) or negative ($h < 1$). For positive cooperativity, lower cooperativity is associated with smaller values of h toward 1. For negative cooperativity, lower cooperativity is associated with larger values of h toward 1.

Experimentally, for long-chain ligands, we observed both positive cooperativity (e.g., CTAB/CTAC/CTAOH) and negative cooperativity (e.g., PVP series). The adsorption cooperativity fundamentally arises from adsorbate-adsorbate interactions on the surface, which can be influenced by the charge, hydrophilicity of the chain as well as the surrounding environment such as the solvent. Therefore, we cannot predict the magnitude or the sign of cooperativity of long chains just based on the surface curvature.

The prediction we can make based on our work is the anti-correlation between affinity and cooperativity: strong adsorbate-surface interactions (larger K) are often associated with weaker cooperativity.

3. Is there an underlying assumption that resazurin adsorption is independent of the facet type? A related question: do different facets have different optimal resazurin concentrations for COMPEITS measurements?

[Reply 6] We did *not* assume that resazurin adsorption is independent of the facet type. For our competition titration (either bulk, single-particle, or sub-particle level), the titration of a range of resazurin concentrations is always an integral part, as we stated briefly on page 12 and in more detail in Supplementary Information section 1.7 of the experimental section. In fact, resazurin adsorption, although showing no cooperativity (Supplementary Information section 1.6), does have facet-dependent adsorption affinity (Supplementary Table 2); we just did not present it in the main text as it was not the focus of this manuscript.

Regarding resazurin concentration for COMPEITS, there isn't an optimal concentration, but a higher concentration is preferred to have a higher rate of the fluorogenic reaction to start with, which will give more frequent detection of the fluorescent product, decreasing the experimental time needed to accumulate statistics for generating the super-resolution images. However, resazurin does have weak fluorescence, so too high a concentration would generate high background, impairing single-molecule detection of the product resorufin. 0.2 μM of resazurin was high enough for all samples for single-molecule imaging in this work while still maintaining a low background.

4. The correlation between h and K is weaker for $h > 1$ and stronger for $h < 1$ (Figs. 1i & 1j). Is there an explanation for why? Line 90-92 points out the correlations, but the correlations are not on equal footing.

[Reply 7] The sign of Pearson's correlation coefficient is rationalizable by that stronger adsorption affinity is associated with weaker cooperativity for both cases. We do not know how to explain the strength of the correlation (i.e., magnitude of the correlation coefficient).

5. The error bars in Figs. 1f & 1g are substantially narrower than the distribution of measurements in Figs. 1i & 1j (also in Fig. 2). It is noted that one is s.e.m. and the other is std. However, the difference is confusing and misleading.

[Reply 8] The values of the bar plots in Fig. 1f or 1g are averages of many particles, for example, from 50 particles for CTAB. Each error bar is the standard error of the mean (s.e.m., which is the standard deviation (s.d.) divided by the number of particles). As the number of particles is quite large, s.e.m. is substantially smaller than s.d.. We chose to use s.e.m. here because the objective was to compare the mean values among different ligands, rather than to focus on the spread (reflected by s.d.) among individual particles.

In contrast, each data point in the scatter plots in Fig. 1i or 1j is from one single particle, and the error bar is the s.d. of the fitted parameter in analyzing the single-particle titration curve. Besides, s.e.m. is not applicable here.

The same reasoning applies to related panels in Fig. 2.

To avoid potential confusions, we added the notes in Fig. 1 caption: "Error bars are s.e.m. in **b-h** for comparing the mean values and s.d. in **i-j** to show the uncertainty of the fitted parameters".

6. How narrow is the pH range where resorufin is brightly fluorescent? Could the pH conformation dependence of PVP be explored by going outside of the optimal range for resorufin?

[Reply 9] Resorufin has a pKa of 5.8 (*Anal Biochem.* **2010**, 399, 7). A basic condition can ensure resorufin to be in the deprotonated fluorescent form. All COMPEITS titrations here were performed in a buffered pH 7.3 solution, in which 97% of resorufin is deprotonated.

At pH 7.3, the repeating units of PVP are fully protonated (*N*-alkylpyrrolidone has a pK_b of ~3.5 and its conjugate acid has a pK_a of ~10.5). Therefore, going to more acidic conditions should not make a significant difference. Going toward more basic conditions (e.g., high than pH 10.5) could start deprotonating the side chains, decreasing the amount of positive charges and leading to conformational changes. This would be an interesting direction for the future, which we have not yet explored. Thanks for the suggestion.

7. How is the optimal resazurin concentration determined? Does it change for particle geometry or ligand selection?

[Reply 10] As we described in **Reply 6** above, there isn't an optimal resazurin concentration but a higher concentration is preferred typically to have higher reaction rates, as long as the background can be kept low.

8. Were the commercial 5 nm Au NPs also coated in silica? Were the smaller particles also imaged with SEM for to check for clustering?

[Reply 11] The 5-nm Au NPs were not coated in silica. They were not imaged with SEM as 5 nm was approaching the resolution limit of the SEM instrument we used. However, we checked the sample under TEM and saw no clustering. At the low density of particles we dispersed on the slide (many microns apart between individual particles), the clustering is even less likely. Even if there is some clustering, the individual particles would be spatially resolvable in super-resolution imaging if they are more than 10-20 nm apart. If they are too close to be resolvable, the measured *K* and *h* are simply averages of the clustered particles.

We added the following statements in Supplementary Information section 3.1: “We optimized the amount of 5-nm Au nanoparticles to be drop casted onto the quartz slide of the flow cell to have low density and ensure minimal clustering of particles. For rare occurrences of particle clustering that are not resolvable at ~10 nm resolution, the measured affinity and cooperativity are the averages of the clustered particles.”

9. The arguments about cooperativity with the silica in lines 161-167: where is the evidence is does not alter behavior? Yes, there is evidence that cooperativity occurs, but why can it not be different than for the small particles (with high surface curvature)? Is *K_L* at least affect by the reduced surface area?

[Reply 12] The presence of mSiO₂ shell could somewhat affect the quantitative values of *h* (i.e., extent of cooperativity), but this shell should not *render* the cooperativity, i.e., cause the adsorption to be cooperative or not, because: (1) the cooperative adsorption for CTAB (and PVP) occurs both in the absence (5-nm Au particles) and in the presence (nanoplates and nanorods) of the shell, and (2) the magnitudes of *h* on the three types of particles are all roughly 2 (or roughly 0.7 for PVP).

The presence of mSiO₂ shell also should not *bias the trend* of the cooperativity to be stronger or weaker, because: (1) we persistently observed the anti-correlation of affinity and cooperativity for CTAB/PVP adsorption on 5-nm Au (without a shell) and nanoplates/nanorods (with a shell), and (2) the {111} facet shows stronger cooperativity than the {110} facet, regardless of whether the {111} facet is located dominantly at low curvature regions (i.e., the top flat facet on nanoplates) or at high curvature regions (i.e., at the tips of nanorods).

Please note that *K_L* is a per-site property, so *solely* reducing surface area (e.g., coverage of some Au surface by the mSiO₂ shell) should have minimal effects. The differences in *K_L* among different regions or types of particles should come from other reasons, such as underlying surface structure, different surface energies, etc.

We expect that for all ligands studied in this work, only the adsorption of PVP might be partially hampered by the mSiO₂ shell, whose mesopores may constrain that only a portion of the PVP chain lands on the Au surface. Nevertheless, adsorption of a polymer chain on a naked, unblocked surface does not guarantee full contact of the entire chain on the surface; in fact, most of the segments could still reside in loops away from the surface, as illustrated in *Macromolecules* **2006**, 39, 6565-6573 and *Macromolecules* **1996**, 29, 7261-7268.

We have added the above discussion about the mSiO₂ shell in the main text (page 6) and in Supplementary Information section 1.2.3.

10. It would be useful to see the raw super-resolution images for individual nanoparticles. The COMPEITS images appear to be binned, but high-resolution images of both [L] = 0 and [L] > 0 would be informative (using marker transparency to

indicate localization density). If claiming 10 nm localization resolution, it would be good to see un-binned results. Also, for fluorescence that appears in multiple frames, it would be interesting to see their distribution of localizations over time.

[Reply 13] We have added exemplary scatter plots of the un-binned product molecule localizations for 5-nm Au particles into Supplementary Fig. 10l-m to show the 10-nm resolution. We have also added the exemplary raw super-resolution images of Au nanoplates and nanorods at $[L] = 0$ and $[L] > 0$ in Supplementary Fig 13g-h and Supplementary Fig 18h-i.

The figure below (added as Supplementary Figure 10n) shows the distribution of localizations for a resorufin molecule lasting for 19 frames on a 5-nm Au particle – all localizations are confined within $\sim 10 \times 10 \text{ nm}^2$, consistent with the resolution.

11. In flow cells, there can be regions of low/no flow due to channeling. If titrating multiple ligand concentrations during a measurement series, how can one be sure there is nothing left from the previous titration? Are the flow cells evacuated between concentrations? There could be a distribution of concentrations across the FOV. Although the individual rate curves suggest this is not an issue due to their high-fidelity fits.

[Reply 14] There are indeed regions of lower flow rates in a flow cell, especially the corners of the flow channel. Our imaging flow channel is $100 \mu\text{m}$ (height) $\times 5 \text{ cm}$ (length) $\times 1 \text{ cm}$ (width) in dimension, and we only chose FOVs of $\sim 60 \times 100 \mu\text{m}^2$ in the center, where the flow rate is expected to follow the designated flow rate and the concentration is expected to be homogeneously distributed in such a small FOV. One certainly cannot guarantee that nothing is left from the previous titration. Therefore, to avoid cross-contaminations from different ligands, one flow cell was only used for one type of ligand, and the titration followed the order from resazurin to the target ligand and from low to high concentrations. The flow cells were not evacuated between concentrations. The later solutions always contained more components or higher concentrations than the previous solution, so the residues from the previous condition would not matter.

12. The frequent use of hyperbolic terms such as "unprecedented" (P5, P9) and "first-of-its-kind" (abs., P3, P8, P16) are, at worst, untrue exaggerations and, at the least, simply unnecessary and distracting. Please respect the intelligence of your reader to understand the significance of your results. It is a misconception that such phrases actually improve your rate of manuscript acceptance.

[Reply 15] We used such terms to help with the initial assessment and planned to remove them before the final version. We have now removed them in the revised manuscript.

13. Is fluorescence from an edge or corner region at least partially physically obstructed? With the high-NA object, light will be collected given the NP sizes, but is there evidence of lower fluorescence intensities near corners and edges?

[Reply 16] To answer these questions, we added the histograms of PSF intensities of resorufin molecules detected at the corner, edge, and flat facet regions, respectively, as Supplementary Fig. 5k. The data show no discernible difference in fluorescence intensity at different regions.

Supplementary Fig. 5 | k, Histograms of PSF intensities of product molecules detected in the corner, edge, and flat facet regions, and the averages and s.d. are 678 ± 308 at the corner region, 710 ± 354 at the edge region, and $689 \pm$

323 at the flat facet region. The PSF intensities are essentially the same across the different regions, indicating there is no spatial bias in the detection of the products of the fluorogenic auxiliary reaction.

14. Within a geometric feature (face, edge), is there is mix of facet types that can be predicted or otherwise determined? For example, on a face, Au NPs are not necessarily atomically smooth. Do steps generate mixtures of facet types and can the ratio of the facet types be estimated?

[Reply 17] The assignments of facet types in this work were based on the electrochemical underpotential deposition (UPD) of Pb monitored by cyclic voltammetry measurements (Supplementary Information section 1.4). We only resolve features corresponding to {111} and {110}, and features belonging to other facets are indiscernible. Consequently, the mixtures of facet types generated by steps, if existing, should not make a major contribution to the measured facet compositions. The results also agreed with the literature (see discussions and references in Supporting Information section 1.2 and 1.3). It is generally accepted that only one type of facet exists within a geometric feature (e.g., {111} for the top facet of a nanoplate) along with defects (could be adatoms, steps, and kinks, etc).

Currently, quantifying the number of defects is a grand challenge, especially on nanoscale particles.

15. The clustering of data in Fig. 3a well-separates the different features. Does Fig. 2h have similar quality clusters if similarly scaled and outliers/tails of the distribution are ignored?

[Reply 18] Fig. 2h did not exist in the original submitted version. Do you mean to compare Fig. 3a with Fig. 2a? If so, data from nanoplates are generally noisier and have less distinction of different regions in the COMPEITS images than those of nanorods. Differences from different regions of nanoplates are better examined by the entire titration range of [L], from which K and h are extracted, instead of just one image generated from only two [L] conditions.

16. Do the geometry differences from synthesis with varying [CTAB] stem from initial seeding of the growth or is it a result of ligand adsorption throughout the synthesis time? Could a synthesis seeded with nanorods dominated by {110} be converted into nanoplates dominated by {111}? Insufficiently exploring the synthesis characteristics is another reason why this part of the study appears an after-thought.

[Reply 19] Please see **Reply 2** above on how the synthesis component of this study was rationalized on the basis of our quantitative COMPEITS imaging results and of past literature.

Thank you also for raising an excellent point on seeding. We think the geometry differences from synthesis with varying [CTAB] stem from ligand adsorption throughout the synthesis time rather than from the initial seeding. Our synthesis was a one-pot approach and does not involve the use of pre-formed seeds, but we understand that nuclei, also called seeds, could still be in situ generated during the nucleation process of a one-pot synthesis. The type of seeds could indeed affect the shape taken by a product particle because the internal structure (e.g., single-crystal vs. twinned structure) could somewhat constrain the shape expression of nanocrystals. However, for nanocrystals growing from the same seeds (i.e., same internal structure), they can still be diverse in shapes depending on the properties of a capping agent or facet directing reagent. For example, single-crystal seeds can grow into cubes and octahedra; penta-twinned seeds can grow into decahedra and nanorods; planer-defect seeds can grow into nanoplates and nanocubes (*Chem. Rev.* **2021**, *121*, 649-735). All these examples of distinctive pairs of particle products are characterized by both different shapes and different facets, despite the same internal structure. Therefore, we think that the “crossover behavior” in the facet distribution of our synthesis with varying [CTAB] should stem from the ligand adsorption rather than the seeding effect. We have added this discussion on seeding in Supplementary Information section 6.8.

Regarding the growth of nanorod seeds into nanoplates, it should not in principle happen because nanorods and nanoplates feature different internal structures. Specifically, nanorods have a penta-twinned structure while nanoplates possess a planar-defect structure.

17. It is unclear if “defect sites” are necessary for COMPEITS (or adhesion in general) or simply provide regions of enhanced activity. The authors casually mention defect density in explanations, but do not define what a defect site is. Are defects regions where the crystalline structure is abruptly inhomogeneous? In that case, does that inaccurately probe the surface in those regions, making this technique sensitive to heterogeneities and not the facet types?

[Reply 20] Defect sites are not necessary for COMPEITS, which only requires the surface sites have a finite affinity toward the ligands leading to adsorption. Yes, in our results defects provide regions of enhanced adsorption affinity. We cannot identify the structural nature of the defect sites based on our technique. In this work, a defect site is a general term for

surface sites that have under-coordinated atoms compared with general facet surface atoms; we expect that both point defects (such as adatoms and vacancies) and line defects are possible candidates.

We did not observe abrupt inhomogeneity in our current or past studies. In our previous works (*J. Am. Chem. Soc.* **2013**, *135*, 1845-1852, and *Nat. Nanotechnol.* **2012**, *7*, 237–241), we discovered that the density of defects on the top facet of nanoplates and on the sides of nanorods shows a gradient from the center of the nanoplates/nanorods toward the periphery/tips, attributable to their changing growth rate during their seeded syntheses.

In SI Section 5.4, we discussed the role of under-coordinated atoms. No matter whether the {110} facets reside on the corners/edges of nanoplates (with more under-coordinated atoms) or on the sides of nanorods (with fewer under-coordinated atoms), they show a larger K and smaller h compared with {111} facets. Therefore, the underlying facets were considered as the main structural characteristics for ligand adsorption at different regions, between which the under-coordinated atoms have less significant contributions. Of course, the differences of K and h in *sub-facet* regions, e.g., inner/middle/outer regions within the *same* flat facets of a nanoplate, are attributed to the differences in density of structural defects which are under-coordinated atoms compared with the regular facet atoms.

We expanded our statement in Line 176-178 to clarify the meaning of defects, which now reads: “We previously established that on the flat {111} facet, the structural defects (i.e., under-coordinated atoms compared with the regular facet atoms) decrease in density from the center toward the periphery because of their seeded growth mechanism.”

18. From previous works (Ref. 30), the defect density variations across a nanoparticle seems to be around 50%. Can the authors speak to expected defect density changes from i/m/o? Is there a length scale from the center of a nanoparticle where the inner region is constant and only deviates within a characteristic distance from an edge?

[Reply 21] Unfortunately, we do not see a way currently. We can only estimate the relative trend but not absolute defect density. The variation in our previous work is surface reactivity that is related to the underlying defect density. Quantifying the actual defect density is a grand challenge. We cannot tell the absolute defect density changes from i/m/o, either, because we do not know the parameters (e.g. reaction rates or adsorption affinities) at the baseline where there is no defect. There is no length scale or characteristic distance because the defect density follows an approximate linear gradient.

19. Although the COMPEITS method does not necessarily require precise super-resolution localization methods since the detection rate is the important component, is it surprising the decisions the authors made for their localization analysis. The background subtraction method is ad hoc (rather than including background emission rates in the fitting term “b”) and inaccurately represents noise in the images, particularly in low-intensity pixels. However, because the authors go through the effort to claim high localization precision in Section S1.8, I must object to their approach.

[Reply 22] Yes, our background subtraction algorithm is not perfect, and the localization precision is not perfect either. However, our claimed “down to ~10 nm” resolution came from localizing the reaction products on single 5-nm Au nanoparticles, whose super-resolution image has a FWHM of ~10 nm (Supplementary Fig. 10b), giving the effective ~10 nm resolution. Please note this ~10 nm resolution is our best scenario, and our typical resolution is about ~10-40 nm (Supplementary Fig. 5d-e) on nanoplates/nanorods, limited by the S/N ratio of single-molecule fluorescence imaging. Also, we interpret our results well within the confidence of our spatial resolution, i.e., all regions we dissect the nanoplates or nanorods are significantly larger than our ~10-40 nm spatial resolution. Please also note that our overall resolution/precision is also affected by correlating optical imaging and electron microscopy, in which the localization error from super-resolution fluorescence imaging was just one out of three components of the overall localization errors of ~40 nm (Supplementary Table 1).

It appears in this section that authors incorrectly apply Eqn. S18 in Eqn. S19. The former produces an estimate of the photon flux impinging on the camera. However, the latter, used to determine localization precision, is based on the number of DETECTED photons. Photons that are not detected cannot improve localization precision and applying these equations as they appear artificially improves the author’s precision estimate.

[Reply 23] Thank you for your carefulness in catching this issue. Indeed, the symbol N in Eq. S18 and Eq. S19 represents the number of photons impinging on the camera and the number of photons detected, respectively. However, the ratio of these two numbers is the quantum yield, i.e., QE in Eq. S18, which has a value of 95-97% for our camera in our detection wavelength range (550-610 nm). The difference in N would only be ~3-5%, and the effect on the localization error is even smaller and negligible.

Nevertheless, we added a note below Eq. S19 to clarify this issue as follows: “The symbol N in Eq. S18 and Eq. S19 represents the number of photons impinging on the camera and the number of photons detected, respectively. However, the

ratio of these two numbers is the quantum yield, i.e., QE in Eqn. S18, which has a value of 95-97% for our camera in the fluorescence detection spectral region (550 – 610 nm). The difference in N is only ~3-5%, and the effect on the localization error Err_i is even smaller and negligible.”

Furthermore, the citations here (Refs. 35 & 54) are references to the author’s previous works. However, the derivation of the precision equation was presented in other works and should be cited correctly (i.e., “Precise Nanometer Localization Analysis for Individual Fluorescent Probes” by Thompson, et al., 2002). It is frustrating that the authors seem to have chosen to cite their own work more prolifically than foundational papers.

[Reply 24] We should have cited the original reference and now have added it as Ref #26 in the Supplementary Information.

20. Despite my objections to the treatment of the background emission, it is also odd that the authors take care to accommodate the sub-pixel shifts of the background images, but still use a Gaussian PSF instead of a pixel-integrated PSF. Why? This has ramifications for the estimated intensities, localization, and the precision analysis. It can also lead a optimization algorithm to erroneous results (such as the PSF width) even for a true detection event.

[Reply 25] Gaussian PSF is commonly used in the literature as a good approximation of the actual PSF, such as in the following references:

- Betzig, E.; Patterson, G. H.; Sougrat, R.; Lindwasser, O. W.; Olenych, S.; Bonifacino, J. S.; Davidson, M. W.; Lippincott-Schwartz, J.; Hess, H. F. Imaging Intracellular Fluorescent Proteins at Nanometer Resolution. *Science* **2006**, *313*, 1642-1645.
- Hess, S. T.; Girirajan, T. P. K.; Mason, M. D. Ultra-High Resolution Imaging by Fluorescence Photoactivation Localization Microscopy. *Biophys. J.* **2006**, *91*, 4258-4272.

We are not entirely sure whether pixel-integrated PSF means the following equation:

$$I(x, y) = A + Bx + Cy + \int_{x-\delta}^{x+\delta} dX \int_{y-\delta}^{y+\delta} dY I_0 \exp \left[-\frac{1}{2} \left(\frac{X - x_0}{\sigma_x} \right)^2 - \frac{1}{2} \left(\frac{Y - y_0}{\sigma_y} \right)^2 \right]$$

where $I(x,y)$ is the intensity counts of the fluorescent molecule in the image at position (x,y) , $A+Bx+Cy$ is a sloping plane to account for the background in the fitting, $I_0 \exp \left[-\frac{1}{2} \left(\frac{X-x_0}{\sigma_x} \right)^2 - \frac{1}{2} \left(\frac{Y-y_0}{\sigma_y} \right)^2 \right]$ is a two-dimensional Gaussian function, and δ is half of the pixel size. Along x or y axis, the integration over each pixel is done numerically by dividing each pixel into 11 equal segments. (x_0, y_0) gives the center location of the PSF. [Adapted from Zhuang and coworkers, *Nat. Methods* **2006**, *3*, 793-796, and *Science* **2007**, *317*, 1749-1753]. If yes, this was the fitting PSF that we used previously (*Nano Lett.* **2009**, *9*, 3968-3973; *Nature Nanotech.* **2012**, *7*, 237-241 and *J. Am. Chem. Soc.* **2013**, *135*, 1845-1852). We plotted the results obtained using this pixel-integrated PSF in the new Supplementary Fig. 5f-j, which are very similar to the results from the Gaussian PSF presented in Supplementary Fig. 5a-e.

21. The SI indicates how non-events and double-detection events are treated in analysis. It would be useful to see the spatial distribution of the rejection rates to see if there is a bias among c/e/f or i/m/o regions of a nanoparticle. Particularly since localization in the edge regions are susceptible to the noise concerns from the background subtraction described above, the authors should demonstrate the rejection rates of detected events in these regions are unaffected by geometry.

[Reply 26] We plotted the spatial distribution of rejection rates below and included it in Supplementary Fig. 5 in the revised version. The rejection rates are approximately the same among c/e/f and i/m/o regions of a nanoplate. However, even though we strive to measure the reaction rates (affected by the rejection rates) stringently, we would like to note again that the trends of the detection rates at different ligand concentrations determine K and h of the ligand, but not the absolute detection rates, which can be adjusted by the concentration of the reactant resazurin.

Supplementary Fig. 5 | 1, The number of fitted localizations and the overall rejection rates at different regions of nanoplates. Rejections consist of filtering based on σ_x and σ_y (either too small or too big) and diffusing molecules (see Section 1.8.2). The data show that there is no significant difference in the rejection rate across different regions. Data were averaged from 55 nanoplates, $[R] = 0.2 \mu\text{M}$, $[\text{NH}_2\text{OH}] = 1.0 \text{ mM}$, and no ligand. Error bars are s.e.m.

REVIEWERS' COMMENTS

Reviewer #1 (Remarks to the Author):

I am in general satisfied with the answers to my questions/remarks. Only for my first question, even for gold there are different allotropes, so the authors kind of not answer the question. But I understand the data I was asking for are not available and I don't want to delay the publication of the paper, especially as conclusions will not change. For me, the paper can be published in its current form.

Reviewer #3 (Remarks to the Author):

The revisions made to the manuscript "Nanoscale cooperative adsorption for materials control" completely satisfy this reviewer, and I would like to thank the authors for their extensive efforts to address every comment in the rebuttal. The significant number of additions and clarifications was helpful to address the points raised. The explanations in the rebuttal letter demonstrate the authors' comprehensive understanding of COMPETES and their particle system. I have no further questions and recommend this manuscript be accepted in its current form.